# The Frequency of Fast Food Consumption in Relation to Wheeze and Asthma Among Adolescents in Gauteng and North West Provinces, South Africa

**DOI:** 10.3390/ijerph17061994

**Published:** 2020-03-18

**Authors:** Vusumuzi Nkosi, Funzani Rathogwa-Takalani, Kuku Voyi

**Affiliations:** 1Environment and Health Research Unit, South African Medical Research Council, Johannesburg 2094, South Africa; 2School of Health Systems and Public Health, Faculty of Health Sciences, University of Pretoria, Pretoria 0001, South Africa; kuku.voyi@up.ac.za; 3Department of Environmental Health, Faculty of Health Sciences, University of Johannesburg, Johannesburg 2094, South Africa; 4Department of Advanced Nursing Science, School of Health Science, University of Venda, Thohoyandou 0950, South Africa; frathogwa@gmail.com

**Keywords:** asthma, wheeze, fast foods, adolescents, South Africa

## Abstract

The prevalence of asthma and allergic diseases have been on the rise, especially in developing countries due to life-style changes. The study aimed to investigate the association between the frequency of fast food consumption with wheeze and asthma among adolescents. A previously validated self-completed questionnaire from the International Study of Asthma and Allergies in Childhood was used to collect information on demographics, socioeconomic status, house characteristics, adolescent respiratory health and diet. The prevalence of wheeze was 38.2% and of asthma was 16.37% among the adolescents. The results from the adjusted regression analyses indicated that eating fast foods three or more times per week was a statistically significant risk factor for wheeze (OR = 1.60; 95% CI: 1.26–2.03) and asthma (OR = 1.37; 95% CI: 1.04–1.91). The study found an association between eating fast foods three or more times per week and wheeze and asthma among adolescents in South Africa. Unhealthy diet plays a crucial role in respiratory health among adolescents; healthy eating habits are encouraged to reduce the burden of respiratory symptoms and diseases.

## 1. Introduction

Over recent decades, the prevalence of asthma and allergic diseases has adequately increased to pose a public health concern [1,2,3]. Many studies have proposed that the increase in asthma and allergic diseases are attributed to changes in our diets [1,2], while others have linked it to environmental factors such as smoking, seasonal variations and air pollution [4,5,6]. A study conducted in children and adolescents in over 40 countries showed that up to 25% of children and nearly 40% of adolescents consume fast foods frequently [7]. The results of this study are consistent with those of other studies, particularly those conducted in the UK and the USA, that showed a high proportion of fast food consumption amongst adolescents [8,9,10]. Another study highlighted an unexpectedly high prevalence of fast food consumption in other developing countries, including those in Asia and Western Europe.

Studies have shown that people living in developed countries have adopted a Westernized lifestyle [2]. This is a diet that mainly consists of fast foods and is rich in processed foods, fats and sugar. In numerous studies, frequent consumption of fast foods has been linked to obesity, hypertension, diabetes mellitus and heart diseases [11,12]. Recent studies have shown an association between the occurrence of asthma and diet. Furthermore, studies have shown that increased consumption of fast foods negatively affects health and increases asthma and wheeze prevalence [6].

Another study conducted among 6–7-year-old children showed that there was a strong association between sugar consumption and asthma [6,13]. A study conducted in adolescents has suggested that the relationship between fast foods and symptoms of asthma, wheeze and eczema is causal, indicating that certain foods increase or decrease the risk of developing asthma, wheeze and eczema [14]. Traditional diets have been associated with lower risks of noncommunicable diseases. Recent International Study of Asthma and Allergies in Childhood study results have shown that frequent consumption of fish, fruits and vegetables was associated with a lower risk of developing asthma [14]. Other studies have identified the intake of fruits, vegetables and fish as protective factors against childhood asthma, while fast food consumption is a risk factor for the disease [15].

Numerous epidemiological studies have been conducted to investigate the association between dietary habits and the risk of asthma in children [16,17,18]. Despite these findings, a study to investigate the frequency of fast food consumption in relation to wheeze and asthma among adolescents has not been conducted in the northern provinces of South Africa. In this study, we investigated the frequency of fast food consumption in relation to asthma and wheeze using data collected from Gauteng and North West in 2012.

## 2. Materials and Methods 

### 2.1. Procedure

Data for this paper were extracted from the cross-sectional study conducted in 2012 among 13–14-year-old pupils who attended schools located in Gauteng and North West provinces of South Africa. A total of 22 randomly selected schools (primary and secondary) were included in the study. Prior to the commencement of the study, the school principals of each school were contacted to explain the reason for the study and expected outcomes. School principals completed the informed consent form and agreed to specific dates for data collection. All eligible pupils (6000) were given an informed consent form for parent/caregiver/guardian to complete. The permission to conduct the study was obtained from the departments of education in Gauteng and North West provinces where the schools were located. Research ethics clearance was granted by the University of Pretoria Ethics Committee (REC number: 235/2011).

### 2.2. Inclusion Criteria

Schools and pupils that did not grant permission before the start of the fieldwork or showed lack of cooperation were excluded from the study. Figure 1 shows a flow chart of the procedure followed to recruit the study participants and the participation rate. 

### 2.3. Questionnaire

The surveys were conducted during school hours by the pupils in their respective schools during the Life-Orientation period using a previously validated questionnaire in South African provinces such as Limpopo, Western Cape and Gauteng from the ISAAC [19,20,21]. Data were collected using the English version of the ISAAC written questionnaire composed of modules such as wheeze and asthma. The medium of instruction in all the schools that participated in the study was English. According to ISAAC Phase 1 protocol, wheeze and asthma were classified on the basis of positive answers to the following questions:

Wheeze: “Have you had wheeze or whistling in the chest at any time?” 

Asthma: “Have you ever had asthma?” 

### 2.4. Confounders 

Potential confounding variables included were: active smoking by participants (yes/no), ETS exposure at home (yes/no), type of residential cooking/heating fuel (electricity, gas, paraffin, wood/coal), the frequency of trucks passing near the residence (never, seldom, frequently through the day, almost all day), sex (female/male), vigorous physical activities (never/occasionally, 1–2 times/week, ≥3 times/week) and type of house (brick, mud, corrugated iron, or combination). 

### 2.5. Data Analysis

Prevalence of the health outcomes and the proportion were calculated by dividing the number of participants who responded affirmatively to a particular question by the number of questionnaires completed. Crude and adjusted odds ratios (OR) and their 95% confidence intervals (CI) were calculated using the univariate and multiple logistic regression analysis (LRA) to estimate the associations between the frequency of fast food consumption, wheeze and asthma. To obtain the adjusted odds ratios (OR) for the effect of fast food consumption and on the health outcomes, both variables were placed in an initial model. This was followed by the addition of a potential confounder in a stepwise manner, starting with the most significant from the univariate analysis. Each time a new potential confounder was added to the model, if the effect estimated between the health outcome and the frequency of fast food consumption already in the model changed by 5%, then the additional variable was retained in the final multiple LRA. Otherwise, the variable was removed, and a different one was added. At the multiple LRA level, variables were deemed significant if *p*-values were <0.05. 

## 3. Results 

### 3.1. Demographic Characteristics

Table 1 shows the profile of the study participants; there were slightly more males (48.50%) than females (47.93%). The majority (83.72%) of the study participants resided in houses that were made of bricks and the main (82.67%) residential fuel type used for cooking and heating the house was electricity. Almost half (48.97%) of the study participants reported having been exposed to environmental tobacco smoke in their homes, and their residences were located near roads were trucks were always passing (48.37%). A proportion of 15.71% reported eating fast foods occasionally in the past 12 months were; 49.30% ate fast foods once or twice per week, and 29.69% did so three times or more per week. 

### 3.2. Prevalence of Wheeze and Asthma

The prevalence of wheeze and asthma is shown in Table 2. Wheeze had the highest prevalence (38.2%), and that of asthma was 16.37%. 

### 3.3. Univariate Regression Analysis

Eating fast foods three or more times per week was a statistically significant risk factor for wheeze (OR = 1.50; 95% CI: 1.22–1.86) and asthma (OR = 1.35; 95% CI: 1.02–1.79) compared to adolescents without wheeze and asthma (Table 3).

### 3.4. Multiple Regression Analysis

Eating fast foods three or more times per week was a statistically significant risk factor for wheeze (OR = 1.60; 95% CI: 1.26–2.03) and asthma (OR = 1.37; 95% CI: 1.04–1.91) (Table 4). Other statistically significant risk factors were active smoking, ETS exposure at home, vigorous physical activities (1–2 times/week, ≥3 times/week) and cooking/heating fuel (wood or coal); these results are shown in Appendix A. 

## 4. Discussion

The study investigated the association between the frequency of fast food consumption with wheeze and asthma amongst adolescents in North West and Gauteng Provinces in South Africa. The prevalence of wheeze was 38.2%, and the prevalence of asthma was 16.37%. Asthma and wheeze prevalence of this study were higher than those of ISAAC studies conducted among adolescents in Polokwane (10.8%) and Cape town (13.3%) in South Africa [19,22]. 

Fast food consumption is rapidly increasing in all parts of the world, especially in developing countries. A study of fast food consumption conducted in children and adolescents has reported that combined frequent and very frequent fast foods consumption was 63% and 79% in South Korea and South Africa, respectively [7]. In this study, 78.99% of the study participants consumed fast foods more than once per week, compared to 32.20% reported in a cohort study conducted in coastal provinces such as the Eastern and the West Cape provinces of South Africa [23]. This difference may be attributed to the fact that Gauteng and North West provinces have high population density because they are the economic hub of the country. 

In this study, eating fast foods three or more times a week was a significant risk factor for wheeze and asthma. Studies conducted to investigate the association between asthma and wheeze and fast food consumption have produced contradicting results. While others have reported an association between fast food consumption and increased asthma and wheeze prevalence, few studies have reported an insignificant association. A cross-sectional study conducted in Shanghai concluded that more frequent consumption of fast foods and sugared drinks were significantly associated with a reduced risk of asthma. In the same study, it was reported that frequent ice cream consumption was significantly associated with a reduced risk of wheeze; however, more frequent consumption of fast foods was significantly associated with the risk of wheeze [24].

An ecological study showed a significant relationship between McDonalds restaurant and the prevalence of wheeze [25]. In a study conducted in Saudi Arabia, fast food consumption was a significant risk factor to wheeze in children and doubled the risk of asthma in adolescents [26]. Frequent hamburger consumption was not only associated with an increased risk of being overweight but also with an increased risk of wheeze. This could be attributed to the high sodium content in hamburgers which has been shown to be a significant risk factor to wheezy illnesses [7]. Fast foods contain higher saturated fatty acids, trans-fatty acids, sodium, carbohydrates, sugar levels and preservatives. An involvement of higher dietary intake of fast foods in the pathophysiology of asthma and atopic diseases may be plausible because the composition of the ingested fatty acids is known to modulate immune reaction [27]. 

Two cross-sectional studies conducted in adolescents have shown no statistical associations between fast food consumption and asthma and wheeze [28,29]. One explanation is that the association between fast foods consumption and asthma may be confounded by BMI. In this study, BMI information was not collected, and the effect estimates might be confounded by BMI. Strong evidence suggests that frequent fast food intake was a risk factor for current severe asthma [18]. In another study, there was a trend between fast food and soft drink consumption and increased risk of asthma and wheeze [30]. 

A systemic review to clarify the associations between consumption of fast foods and asthma and allergies included 13 cross-sectional studies. Three case-control studies concluded that the consumption of fast foods, mainly hamburgers, correlates with asthma in a dose-response pattern. Most of the studies included in this review showed that, when adjusted for confounding factors such as sex/gender, education level, maternal smoking, smoke exposure, and obesity, there was a strong association between fast food consumption and asthma and allergic diseases [5,18,31,32,33]. A case-control study conducted in adolescents in Saudi Arabia found that the frequency of eating in a fast food outlet was significantly associated with asthma [26]. A multicenter study of nearly 400,000 adolescent study participants showed that fast food consumption might contribute to the increasing prevalence of asthma and rhino-conjunctivitis [14]. Another multicenter study of over 50,000 adolescent study participants indicated that high consumption of burgers and fizzy drinks is associated with an increased lifetime prevalence of asthma [16]. A temporal relationship between fast food consumption and asthma and wheeze is well established from cross-sectional studies. Research studies are required to investigate whether fast food consumption is involved in the development of asthma and its symptoms. 

Some statistically significant risk factors for wheeze included active smoking, ETS exposure at home and vigorous physical activity 1–2 or ≥3 times/week. Cooking and heating using wood or coal were also shown to be significant risk factors. Our study results concur with results from numerous other studies. To the best of our knowledge, no other research has been conducted to assess the relationship between fast food consumption and asthma and wheeze in South Africa.

### Limitations

The major strength of this study was that it used a previously validated questionnaire from ISAAC, which has been used worldwide for asthma and wheeze research. This study had some limitations associated with a cross-sectional study design, as it cannot provide any evidence of causality. The findings are based on self-reported answers from a questionnaire. Self-reported responses may lead to misclassification of the disease and exposure status, which may result in spurious statistically significant associations. Due to logistical challenges during data collection, height and weight of the adolescents were not measured; therefore, BMIs were not calculated. BMI was not included as one of the confounding variables in the analyses. In addition, there is a wide variety of fast foods, and there was no information on the specific food names provided in the questionnaire. Lastly, no atopic tests were conducted on the study participants. 

## 5. Conclusions

The study found an association between eating fast foods three or more times per week and wheeze and asthma among adolescents in South Africa. Unhealthy diets play a crucial role in respiratory health among adolescents; healthy eating habits are encouraged to reduce the burden of respiratory symptoms and diseases.

## Figures and Tables

**Figure 1 ijerph-17-01994-f001:**
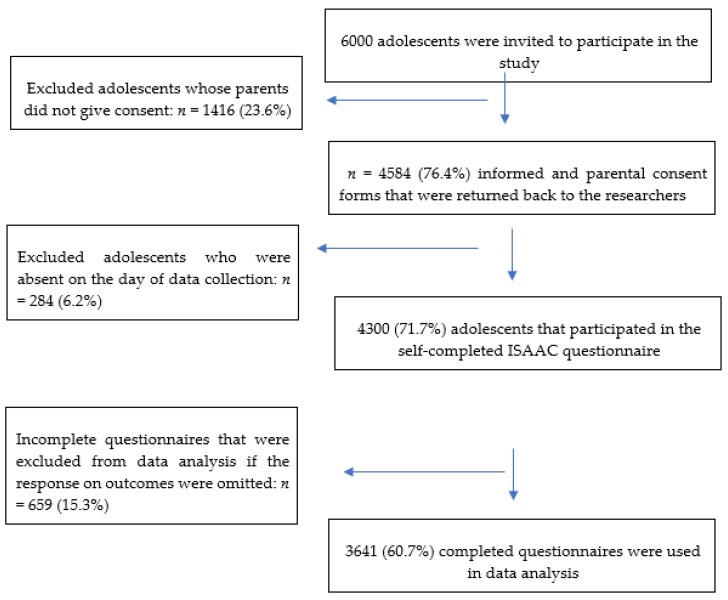
Recruitment procedure of study participants and participation rate.

**Table 1 ijerph-17-01994-t001:** Demographic characteristics of the study participants (*n* = 3641).

Characteristics	Frequency (*n*)	Percentages (%)
**Sex**		
Female	1745	47.93
Male	1766	48.50
Missing	130	3.57
**Type of house**		
Brick	2895	83.72
Mud	73	2.11
Corrugated iron	304	8.35
Combination	102	2.80
Other	84	2.31
Missing	183	5.03
**Residential fuel type used for cooking/heating**		
Electricity	3010	82.67
Gas	93	2.55
Paraffin	139	3.82
Open fires	113	3.10
Missing	286	7.85
**Vigorous physical activity per week**		
Never	1088	29.88
Once or twice per week	1481	40.68
Three times or more per week	950	26.09
Missing	122	3.35
**Frequency of trucks passing near the residence on weekdays**		
Never	556	15.27
Seldom	723	19.86
Frequently	491	13.49
Always	1761	48.37
Missing	110	3.02
**ETS exposure at home in the past 30 days**		
No	1433	39.36
Yes	1747	47.98
Missing	461	12.66
**Do you smoke cigarettes?**		
No	3386	93.00
Yes	151	4.15
Missing	104	2.86
**How often did you eat fast foods in the last 12 months?**		
Never or occasionally	572	15.71
Once or twice per week	1795	49.30
Three or more times per week	1081	29.69
Missing	193	5.30

**Table 2 ijerph-17-01994-t002:** Prevalence of wheeze and asthma among the study participants (*n* = 3641).

Health Outcome	Frequency (*n*)	Percentage (%)
**Wheeze**		
No	2156	59.21
Yes	1391	38.20
Missing	94	2.58
**Asthma**		
No	2813	77.26
Yes	596	16.37
Missing	232	6.37

**Table 3 ijerph-17-01994-t003:** Unadjusted odds ratios of wheeze, asthma and fast foods among the study participants (*n* = 3641).

Wheeze			
How often Did You Eat Fast Foods in the Last 12 Months?	Unadjusted OR	95% CI	*p*-Value
Never or occasionally	1	1	1
Once or twice per week	1.04	0.85–1.27	0.693
Three or more times per week	1.50	1.22–1.86	<0.001
**Asthma**			
Never or occasionally	1	1	1
Once or twice per week	1.05	0.81–1.37	0.706
Three or more times per week	1.35	1.02–1.79	0.032

**Table 4 ijerph-17-01994-t004:** Adjusted odds ratios of wheeze, asthma and fast foods among the study participants (*n* = 3641).

Wheeze ^a^			
How often Did You Eat Fast Foods in the Last 12 Months?	Adjusted OR	95% CI	*p*-Value
Never or occasionally	1	1	1
Once or twice per week	1.09	0.87–1.37	0.434
Three or more times per week	1.60	1.26–2.03	<0.001
**Asthma ^b^**			
Never or occasionally	1	1	1
Once or twice per week	1.05	0.81–1.37	0.706
Three or more times per week	1.37	1.04–1.91	0.050

Note: ^a^ Model adjusted for ETS exposure at home in the past 30 days, residential fuel type used for cooking/heating, type of house, vigorous physical activity per week, cigarette smoking, frequency of trucks passing near the residence on weekdays. ^b^ Model adjusted for ETS exposure at home in the past 30 days, cigarette smoking, vigorous physical activity per week, residential fuel type used for cooking/heating, type of house.

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
