# Peer review of "The Frequency of Fast Food Consumption in Relation to Wheeze and Asthma Among Adolescents in Gauteng and North West Provinces, South Africa"

_ijerph, 2020, doi:10.3390/ijerph17061994_

Round 1

Reviewer 1 Report

This work addresses an important topic of association of fast food consumption and the risks of wheeze and asthma.  There are few limitations which has been addressed in the paper.

Major issues

  • Which foods were considered as fast foods? There is a wide variety of fast foods. If there was no info on the specific food name, it should be included in the limitations.
  • Was BMI affected by fast food consumption? If so, then effect of BMI should be analysed.
  • What was the status of atopy in the study population?
  • Was there any effect of ethnic background on the dietary habit and risk of wheeze and asthma?
  • In the discussion, the authors primarily listed what others have found and did not discuss their own result. The authors need to re-structure the discussion e.g. describe/discuss their own findings and how these align or contradict with others and why?
  • Please add a discussion point on how fast food consumption can affect wheeze and asthma- is it related to immune development or microbiota or other factors?

Minor issues

Line 97: please add reference of ISAAC questionnaire.

Author Response

Review report 1

This work addresses an important topic of association of fast food consumption and the risks of wheeze and asthma.  There are few limitations which has been addressed in the paper.

Thank you for your comment

Major issues

Which foods were considered as fast foods? There is a wide variety of fast foods. If there was no info on the specific food name, it should be included in the limitations.

Thank you for your comment. We agree with the reviewer that there is a wide variety of fast foods. There  was no information on the specific food names provided in the questionnaire. This has been included as one of the study limitations.

Was BMI affected by fast food consumption? If so, then effect of BMI should be analysed.

Thank you for your comment. Due logistical challenges encountered during data collection. The height and weight of the adolescents were not collected; thus, we could not calculate BMI. This is mentioned as one of the study limitation

What was the status of atopy in the study population?

Thank you for your comment. The atopy of the study population is not known. A study is currently being conducted assess atopy among the adolescents.

Was there any effect of ethnic background on the dietary habit and risk of wheeze and asthma?

Thank you for your comment. No effect was observed of ethic background on the dietary habits and the risk of wheeze and asthma.

In the discussion, the authors primarily listed what others have found and did not discuss their own result. The authors need to re-structure the discussion e.g. describe/discuss their own findings and how these align or contradict with others and why?

Thank you for your comment, the discussion section was re-structured in relation to the findings of this study and others.

Please add a discussion point on how fast food consumption can affect wheeze and asthma- is it related to immune development or microbiota or other factors?

Thank you for your comment, this sentence was added “Fast foods contain higher saturated fatty acids, trans fatty acids, sodium, carbohydrates, sugar levels and preservatives. An involvement of higher dietary intake in fast food in the pathophysiology of the asthma and atopic diseases may be plausible because the composition of the ingested fatty acids is known to modulate immune reaction.”

Minor issues

Line 97: please add reference of ISAAC questionnaire.

Thank you for your comment. References were added.

Reviewer 2 Report

This large cross-sectional study (n=3641) showed that the consumption of fast food more than 3 times per week increases the risk for wheeze and weakly the risk for asthma among 13-14 years old pupils in South Africa. Although several studies exist showing a positive link between fast food consumption and allergic diseases, this study is one of the first conducted in South Africa including large sample size. The prevalence of allergies is steadily increasing in African countries and early prevention strategies are needed to attenuate the burden of asthma. Although, this study is of high value since only a limited number of large-scale studies of African countries are available, there are some limitations that need to be addressed.

  1. Are pupils with wheezing symptoms free of asthma and vice versa. Are the controls free of wheeze AND asthma?
  2. Bodyweight is known to be associated with asthma. Is the frequency of fast food consumption linked to children’s BMI? Please include z-score BMI in the models.
  3. Are there gender-dimorphic effects?
  4. Please, discuss the results in the context of reference: DOI: https://doi.org/10.1016/j.jaci.2019.07.048, which is a recently published study examining environmental factors that are associated with allergy in urban and rural children from the South African Food Allergy (SAFFA) cohort. What does the current study add?
  5. Please indicate that ORs are from adjusted regression analyses in the abstract.
  6. There is a doubling in line 105: “Confounders potential confounding variables…”
  7. Line 151: Please, add at the end of the sentence to which control group is a risk increase observed, e.g. “… compared to children without wheeze and/or asthma.”.

Author Response

Review report 2

This large cross-sectional study (n=3641) showed that the consumption of fast food more than 3 times per week increases the risk for wheeze and weakly the risk for asthma among 13-14 years old pupils in South Africa. Although several studies exist showing a positive link between fast food consumption and allergic diseases, this study is one of the first conducted in South Africa including large sample size. The prevalence of allergies is steadily increasing in African countries and early prevention strategies are needed to attenuate the burden of asthma. Although, this study is of high value since only a limited number of large-scale studies of African countries are available, there are some limitations that need to be addressed.

Thank you for your comment.

Are pupils with wheezing symptoms free of asthma and vice versa. Are the controls free of wheeze AND asthma?

Thank you for your comment. The controls were free of wheeze and asthma.

Bodyweight is known to be associated with asthma. Is the frequency of fast food consumption linked to children’s BMI? Please include z-score BMI in the models

Thank you for your comment. Due logistical challenges encountered during data collection. The height and weight of the adolescents were not collected; thus, we could not calculate BMI. This is mentioned as one of the study limitation, “

Are there gender-dimorphic effects

Thank you for your comment. Sex was not found to be a significant risk factor. In lines 168 – 170 in the manuscript, other significant risk factors are mentioned

Please, discuss the results in the context of reference: DOI: https://doi.org/10.1016/j.jaci.2019.07.048, which is a recently published study examining environmental factors that are associated with allergy in urban and rural children from the South African Food Allergy (SAFFA) cohort. What does the current study add?

Thank you for your comment. In the discussion section of the manuscript, the sentence “In this study, 49.30% of the study participants consumed fast foods more than once per week compared to 32.20% reported in cohort study conducted in coastal provinces such as the Eastern and the West Cape provinces of South Africa. This difference may be attributed to the fact that Gauteng and North-West provinces have high population density because they are the economic hub of the country; and adolescents are exposed to different types of environmental factors such as pollution from industries and mining.”

According to the best of our knowledge, this is the first study that has focused in the frequency of the frequency of fast food consumption in relation to asthma and wheeze in the interior provinces of South Africa.

Please indicate that ORs are from adjusted regression analyses in the abstract.

Thank you for your comment. The sentences in the abstract now read “The results from the adjusted regression analyses indicated that easting fast foods three or more times per week was statistically significant risk factor for wheeze (OR = 1.60; 95% CI: 1.26 – 2.03) and asthma (OR = 1.37; 95% CI: 1.04 – 1.91).”

There is a doubling in line 105: “Confounders potential confounding variables…”

Thank you for your comment. The double in the line was fixed. The sentences now read “Potential confounding variables were……….”

Line 151: Please, add at the end of the sentence to which control group is a risk increase observed, e.g. “… compared to children without wheeze and/or asthma.”.

Thank you for your comment. The sentence was added, please lines 152 -153 of the manuscript.

Reviewer 3 Report

Presented study based on large population of students  investigate the association 17 between the frequency of fast foods consumption with wheeze and asthma among adolescents. The health out outcomes - wheezing and asthma were assessed by validated by ISAAC questionnaire. The tools used for exposure - consumption of fast food  meals are not presented. Probably they included some questionnaire. No information is provided about the validation ot his tool.

No information was provided about the quality of meals served at home. Long term diet patterns  Reviewer would like to know the reasons why this factor was not included as confounder?

The maternal diet during pregnancy maybe important as well (see publication of Bédard A. Eur Respir J. 2017 Jul 5;50

The positive aspects of presented study are the clear presentation of results, However more information about the direction of the association  between given confounders and outcomes would be appreciate

Author Response

Review report 3

Presented study based on large population of students investigate the association 17 between the frequency of fast foods consumption with wheeze and asthma among adolescents. The health out outcomes - wheezing and asthma were assessed by validated by ISAAC questionnaire. The tools used for exposure - consumption of fast food meals are not presented. Probably they included some questionnaire.

Thank you for your comment

No information is provided about the validation of his tool.

In the lines 97-99 of the manuscript it is mentioned that “The surveys were conducted during school hours by the pupils’ in their respective schools during the Life-Orientation period using a previously validated questionnaire in South African provinces such as Limpopo, Western Cape and Gauteng from the ISAAC

No information was provided about the quality of meals served at home. Long term diet patterns Reviewer would like to know the reasons why this factor was not included as confounder?

Thank you for the comment. This information was not collected as it is not included in the ISAAC questionnaire.

The maternal diet during pregnancy maybe important as well (see publication of Bédard A. Eur Respir J. 2017 Jul 5;50

Thank you for the comment. Authors agree with the reviewer that the diet of the mother during pregnancy play a significant role in childhood respiratory and atopic outcomes. The information on maternal diet during pregnancy was not collected.

The positive aspects of presented study are the clear presentation of results, However more information about the direction of the association between given confounders and outcomes would be appreciate.

Thank you for the comment. In the manuscript it is stated that “Other statistically significant risk factors were active smoking, ETS exposure at home, vigorous physical activities (1 to 2 times/week, ≥ 3 times/week) and cooking/heating fuel (wood or coal).” The focus of this manuscript was on the association between frequency of fast food consumption, asthma and wheeze.

Round 2

Reviewer 1 Report

Thanks for addressing all the comments. 

Author Response

To the Editor

IJERPH

Manuscript ID: ijerph-732010

13 March 2020

Dear Editor,

Please find enclosed the revised manuscript “The frequency of fast foods consumption in relation to wheeze and asthma among adolescents in Gauteng and North West provinces, South Africa” as well as a point-by-point answer to each of the comments.

Thank you for the opportunity to improve the manuscript.

Best wishes,

Vusumuzi Nkosi, on behalf of all authors

Thanks for addressing all the comments. 

Thank you for your comment.

Reviewer 2 Report

Although the manuscript has been improved, there are still some points that needs to be addressed:

Line 59-60: This study is not the first study conducted in South Africa. Please rephrase the sentence accordingly.

Line 168-170: The authors mentioned that sex was not found to be a significant risk factor, but other factors were related to wheeze/asthma. Please, provide OR to support these statements.

Line 188-195: In the SAFFA cohort differences between the association of fast food consumption and asthma in rural vs. urban children has been identified. Based on these observations, the discussion should be extended.

Line 191: Please, indicate the asthma prevalence of ISAAC studies conducted among adolescents in Polokwane and Cape town in South Africa.

Line 203-204: Why might be the increased frequency of fast food consumption related to environmental pollution?

Line 152: remove „or“

Author Response

To the Editor

IJERPH

Manuscript ID: ijerph-732010

13 March 2020

Dear Editor,

Please find enclosed the revised manuscript “The frequency of fast foods consumption in relation to wheeze and asthma among adolescents in Gauteng and North West provinces, South Africa” as well as a point-by-point answer to each of the comments.

Thank you for the opportunity to improve the manuscript.

Best wishes,

Vusumuzi Nkosi, on behalf of all authors

Line 59-60: This study is not the first study conducted in South Africa. Please rephrase the sentence accordingly.

Thank you for your comment, the sentence was amended accordingly to “ Despite these findings a study to investigate the frequency of fast foods consumption in relation to wheeze and asthma among adolescents has not been conducted in the northern provinces of South Africa”.

Line 168-170: The authors mentioned that sex was not found to be a significant risk factor, but other factors were related to wheeze/asthma. Please, provide OR to support these statements.

Thank you for your comment. The results of other risk factors of asthma and wheeze are shown in the supplementary tables 1 & 2. In the manuscript, it is also mentioned that the results of other risk factors are shown in Supplementary Table 1 and 2.

Line 188-195: In the SAFFA cohort differences between the association of fast food consumption and asthma in rural vs. urban children has been identified. Based on these observations, the discussion should be extended.

Thank you for your comment. The main finding of the SAFFA cohort study is that high fast food consumption (once a week or more) was associated with higher rates of atopic dermatitis in urban children, and higher rates of aeroallergen sensitization in rural children.

In this study, no atopic tests were conducted on the study participants; this is mentioned as one of the study limitations. Based on the reason above, it makes it difficult to expand the discussion by comparing our findings with SAFFA’s cohort, even though our study was conducted in urban settings.

Line 191: Please, indicate the asthma prevalence of ISAAC studies conducted among adolescents in Polokwane and Cape town in South Africa.

Thank you for your comment. The prevalence of asthma in Polokwane and Cape Town is provided in the manuscript.

Line 203-204: Why might be the increased frequency of fast food consumption related to environmental pollution?

Thank you for your comment. After careful consideration, statements regarding environmental pollution were deleted on the manuscript.

Line 152: remove „or“

Thank you for your comment. In line 152 “or” is deleted